# Children from the poor families seem to grow up earlier: An examination of how family economy stress links to career exploration

**Xiao-li Cheng[1], Guang-ya Ma[2], Lu-yao Zhang[3]\*, Lei Lu[4]**

**1** Jinhua Open University, Jinhua, Zhejiang, China, **2** School of Foreign Studies, Yiwu Industrial and Commercial College, Jinhua, China, **3** Suzhou Vocational Health College, Suzhou, Jiangsu, China, **4** School of Psychological and Cognitive Sciences, Beijing Key Laboratory of Behavior and Mental Health, Peking University, Beijing, China

\* 340562427@qq.com

**Data Availability Statement:** All relevant data are within the paper and its Supporting Information files.

## Abstract

Career exploration, a critical antecedent to college students' career choices and employment, offers essential guidance for their career development. However, existing research has not explored how changes in family economic stress impact career exploration. Drawing on social cognitive career theory, this study aims to deepen our understanding of the complex interplay between family economic stress and career exploration among vocational school students. We tested our hypotheses using three-time lagged data collected from 600 vocational school students studying in mainland China. As expected, family economic stress positively predicts career exploration via love of money. Additionally, both students' effective part-time behavior and teacher support were found to positively moderate the indirect relationship between family economic stress and career exploration through love of money. We attempt to highlight the broader implications of understanding the nuanced ways in which economic background shapes career exploration. Theoretical and practical implications are discussed.

## Introduction

The Chinese education department's increasing emphasis on higher education has led to a widespread expansion of university admissions [1–3]. Consequently, the number of university students is growing annually, and the issue of graduate employment is becoming more pronounced [4]. The difficulty of employment for university graduates has become a major focus for both society and researchers. The university phase is a crucial period for developing values and life attitudes, exploring suitable career directions [5], clarifying future professional roles [6], and aiding future career success [7, 8]). Additionally, whether the economic pressures of the post-COVID era have promoted intensified career exploration among college students to alleviate family financial stress remains an unresolved research topic. Family economic stress, defined as the stress experienced when family needs go unmet [9], raises questions about its potential as a motivational force for career exploration [10]. Therefore, this study focuses on

**Funding:** This paper is supported by funding from the third batch of Research and Innovation Team Projects at Zhejiang Open University and Guangdong Province Education Science Planning Project (No: 2021WTSCX255. Research and Practice of Performance Salary System in Higher Vocational Colleges based on Contingency Theory).

**Competing interests:** The authors have declared that no competing interests exist.

vocational college students to examine the spillover effects of family economic stress on their career exploration. It delves into career exploration behavior from three perspectives: the students' families, the students themselves, and universities, attempting to fill the research gap by examining career exploration from multiple angles. The study endeavors to address the critical and intriguing question, "Do children from poor backgrounds seem to grow up earlier?"

Family economic stress is defined as an individual's subjective sense of stress when family resources are unable to meet family needs [9]. The family stress model points out that economic difficulties such as low income, debt and negative financial events cause family economic stress, thereby affecting individual development [11]. For example, Iovu et al. (2018) [12] proposed that material deprivation has a negative impact on students' future expectations. Therefore, most studies discuss the negative impact of college students' financial pressure on personal learning or output, and how to alleviate this negative effect, which is not conducive to the personal development of college students [13].

However, it is worth noting that according to social cognitive career theory (SCCT), individual career development is determined by individual differences, environmental factors and behavior, and the interaction of these three directly affects career goals and career exploration [14]. The family economic status of higher vocational students affects their behavior through individual subjective perception [15]. In terms of environmental factors, the behavior of college students is mainly affected by family and university [16]. In terms of family, the financial pressure of the family cause students to feel a sense of financial deprivation and be unable to meet their own financial needs in life and study [17], thus creating a sense of love of money [18]. Love of money affirms the positive attributes of money and believes that money is good and an important source of respect from others [19]. It emphasizes the fairness of money acquisition and that money has a strong motivational effect on them [20]. From the perspective of college students themselves, college students who worship money will show a strong motivation to make money and proactively explore job hunting in order to satisfy their own sense of love of money and relieve family economic pressure.

In terms of colleges and universities, student behavior is mainly affected by teacher support [21]. College students perceive that teachers provide support, which has a positive effect on college students' attitudes and behaviors [22]. Teacher support gives students more encouragement and resources in life and study, promotes college students' career exploration behavior, and helps them build their careers [23]. Still, according to the social cognitive career theory [14], one's own differentiated behaviors will affect career development, and effective part-time behavior will help optimize career exploration [24, 25]. Thus, this study investigates the complex dynamics between family economic stress and career exploration among vocational school students, focusing on how financial challenges in early life can accelerate maturity and career-related decisions. This study draws from the social cognitive career theory (SCCT), which posits that individual career development is a product of the interplay between personal differences, environmental factors, and behavior. This study's unique contribution lies in examining the mediating role of love of money and how it bridges the gap between economic hardships and proactive career exploration, a perspective that is relatively unexplored in existing literature.

In conclusion, this study highlights the broader implications of understanding the nuanced ways in which economic background shapes career exploration among vocational school students in China. Based on the social cognitive career theory (SCCT), this study discusses the predictive effect of family economic stress on career exploration, and introduces love of money as the mediating variable to explain the influence path of this predictive effect. At the same time, considering the intervention effect of vocational students' teacher support and effective part-time behavior on career exploration, we seek to explain the formation

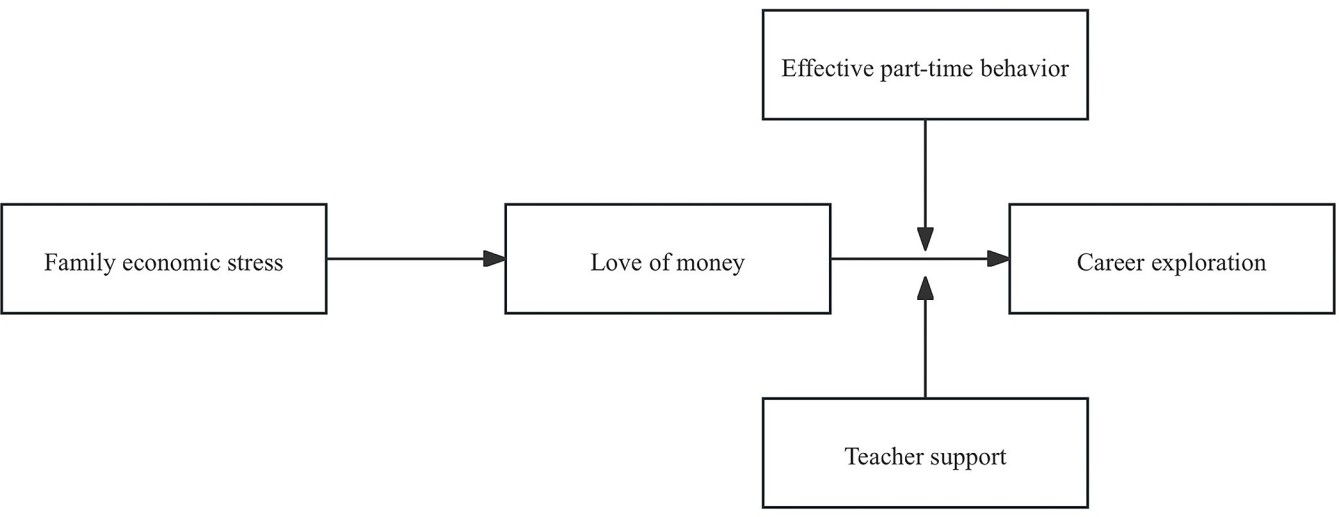

**Fig 1. Hypothesized research model.**

mechanism of career exploration from the family environment, university factors and their own differential behavior. It underscores the importance of supportive structures, such as work opportunities and educational guidance, in mitigating the adverse effects of family economic stress and guiding youths towards constructive career pathways, which buffer the impact of economic stress on career decision-making processes. Fig 1 shows our research model.

## Theory and hypothesis development

### Family economic stress and career exploration

Family, as a crucial environment for individual growth and socialization [26], significantly influences college students' academic and career development [27], especially under economic stress. Economic stress, understood as the difficulty in meeting needs with available family resources, has a profound impact on students' career choices and construction [9]. According to the social cognitive career theory [14], college students' family environment factors have a differentiated impact on their job search and career construction [28]. Firstly, economic stress often leads students to prioritize practical and financially stable career paths, providing immediate economic relief to their families [29]. Secondly, students from economically stressed backgrounds typically gain workforce exposure earlier, enhancing their understanding of diverse career options [30]. Economic hardship can also serve as a motivator, prompting students to actively explore a range of career opportunities to achieve better financial security [31, 32]. Thirdly, students facing financial challenges tend to value and more deeply engage in career development opportunities [33]. Due to economic pressure, they are unable to meet their material and spiritual needs, which leads them to actively and eagerly seek job opportunities in order to obtain the benefits generated by work. Finally, the quest for financial independence drives students to consider careers offering rapid and stable financial returns [10, 34]. The desire for financial independence is a powerful motivator for students when selecting their career paths. This drive often leads them to prioritize careers that are known for rapid and consistent financial returns. Students facing financial pressures or aspiring to achieve financial autonomy as soon as possible are more likely to gravitate towards careers that offer immediate and stable income. This approach enables them to quickly address personal or

family financial needs and establish a secure financial foundation. Consequently, students may lean towards professions with a clear and reliable pathway to financial success, rather than those that might offer slower financial growth or less certainty in terms of income. Therefore, hypothesis 1 is proposed:

**Hypothesis 1**: *Family economic stress positively predicts vocational students' career exploration.*

## Love of money as a mediator

When students face family economic stress, their focus often shifts towards ensuring financial stability [35]. This shift is not just a matter of preference but becomes a necessity, influencing their career choices. The financial pressure experienced at home can lead to a heightened 'Love of money' or a stronger desire for financial security, which in turn significantly influences their career exploration [19]. Students might prioritize careers that promise more immediate and stable financial rewards over those that align with their personal interests or talents [36]. This behavior is a pragmatic response to their economic circumstances, where financial stability takes precedence in their career decision-making process [37]. Essentially, the economic challenges faced at home shape their professional aspirations and choices, emphasizing the practical need for financial security in their future careers.

Love of money conceptualized as (1) one's attitudes towards money; (2) one's meaning of money; and (3) one's wants, desires, values and aspirations of money; it is divided into three dimensions: rich, motivator and important [19, 38] First, students from economically stressed families might view achieving wealth as a primary goal [39]. This aspiration for financial richness guides their career exploration towards fields that promise higher earnings. Second, the desire to improve one's financial situation can be a powerful motivator [40, 41]. For students experiencing family economic stress, the pursuit of wealth becomes a driving force in their career decision-making process. Financial considerations often take precedence in career choices for these students [42]. The importance of financial security and the desire to attain wealth become key factors in their exploration of potential career paths [38, 40].

**Hypothesis 2**: *Love of money mediates the relationship between family economic stress and career exploration.*

## Teacher support as a moderator

The social cognitive career theory posits that individuals are not isolated entities, and their choices are shaped by both internal factors and the external environment [28, 43]. Specifically, in environments characterized by positive support, individuals are motivated to actively pursue their goals [28]. During the transition from the student role in school to the professional status, school and teachers play a pivotal role in influencing the career development of college students [22]. Teachers' support yields a positive impact on students' confidence, attitudes, and behaviors [44]. Perceived teacher support refers to students' perception of the support they receive from teachers regarding their learning, attitudes, and abilities. In the context of career exploration, teachers stimulate external factors that guide college students, offering adequate support and encouragement to mitigate deviant behaviors [23].

The presence of supportive teachers fosters an environment in which students feel more confident and guided in their career decisions, particularly when financial incentives influence these decisions. Teachers play a dual role, offering practical advice and emotional support, assisting students in navigating the intricate interplay between financial considerations and career aspirations [45, 46]. Within a supportive educational context, students are more likely

to discover meaningful and satisfying ways to reconcile their desire for financial gain with their exploration of career paths [47, 48], resulting in more informed and balanced career choices [49]. Firstly, teachers can direct students in aligning their financial motivations with suitable career paths, thereby enhancing the efficacy of their career exploration [50]. Teacher support is instrumental in aiding students to set pragmatic career goals that strike a balance between financial aspirations and personal skills and interests. Secondly, supportive teachers can alleviate stress associated with financial concerns, enabling students to explore career options more freely [51]. Thirdly, teachers can provide resources and opportunities for students to explore careers that fulfill both their financial goals and professional interests [46, 52]. In instances where college students grapple with conflicting and negative emotions while facing financial pressure and exploring job opportunities, emotional support from teachers empowers students to confidently pursue careers that align with their financial needs. Finally, teachers can act as role models, exemplifying how to integrate financial aspirations with professional satisfaction (Son & Kim, 2021). Therefore, Hypothesis 3 is proposed:

**Hypothesis 3:** *Teacher support positively moderates the positive relationship between love of money and career exploration. The higher the level of teacher support, the stronger the positive relationship between love of money and career exploration; otherwise, the weaker the relationship.*

## Effective part-time behavior as a moderator

Engaging in part-time employment serves as a beneficial strategy for college students to gain insights into various jobs, clarify future career paths, and enhance work adaptability before formally entering the workforce [24, 25]. The effectiveness of part-time behavior in college students represents a value-conditioned resource that is intricately linked to future job development and career success [46]. Such effective part-time work behavior facilitates the transition from exploring potential careers to developing adaptability for future employment, particularly when aligned with clear career goals. This involvement promotes the development of a distinct career identity, intensifies the exploration of job opportunities, and ultimately contributes to the acquisition of job adaptability [53].

Effective part-time engagement serves as a pivotal link between a student's financial objectives and their career exploration [54]. Students who manage part-time employment effectively gain a practical understanding of how various careers can help them achieve their financial goals. This direct exposure to the workforce empowers them to align their career exploration more effectively with their financial motivations. The skills, confidence, and financial acumen acquired from part-time positions empower students to make well-informed decisions regarding their future careers, particularly in terms of financial viability and stability [55]. Firstly, part-time work exposes students to various job roles and industries, broadening their comprehension of the career landscape [56]. In certain instances, engaging in part-time employment provides hands-on experience, aiding students in comprehending the real-world implications of different careers, particularly in financial terms [57]. Secondly, students develop skills and work ethics through part-time employment, which proves valuable in their overall career exploration [58]. Effective part-time behavior assists students in striking a balance between their financial needs and career aspirations, leading to more well-informed career decisions. Lastly, part-time work enhances financial literacy, making students more cognizant of the financial aspects associated with various careers [56]. This strengthens the connection between financial motivations and career exploration [59]. Successful engagement in part-time work builds confidence, motivating students to explore careers that align with their financial goals [60, 61]. Therefore, Hypothesis 4 is proposed:

**Hypothesis 4:** *Effective part-time behavior positively moderates the positive relationship between love of money and career exploration. The higher the level of effective part-time behavior, the stronger the positive relationship between love of money and career exploration; otherwise, the weaker the relationship.*

## A moderated mediation model

This study introduces a moderated mediation model that integrates the mediating effect outlined in Hypothesis 2 and the moderating influences of Hypothesis 3 and Hypothesis 4. Economic stress experienced by vocational students within their families prompts them to engage in career exploration, aiming to mitigate the spillover impact of economic pressure. Specifically, students' affinity for financial gain acts as a mediating factor between family economic stress and career exploration. However, the mediating influence is influenced by students' perceived teacher support and effective part-time behavior. Elevated levels of teacher support and effective part-time behavior among students enhance the positive indirect effects of family economic stress and career exploration. Consequently, Hypothesis 5 is posited:

**Hypothesis 5**: *Both vocational student perceived teacher support and effective part- time behavior positively moderate the indirect effect of family economic stress on career exploration through love of money. When student have a higher level of perceived teacher support and effective part- time behavior, this indirect effect is stronger; on the contrary, this indirect effect is weaker.*

## Method

### Participants and procedure

According to the career stage division, it is generally believed that young adults in their twenties are in the exploratory period, developing career-related cognition. Therefore, this study selects college students in the career-choosing period as the research subjects, as they represent a critical stage in career development. The sample included 12 vocational colleges from Beijing, Shanghai, Guangdong Province, Jiangsu Province, and Zhejiang Province. These institutions were chosen for their diverse student bodies from across the country, ensuring sample diversity.

Before the formal survey, teachers and students involved in the study communicated with participants to clarify any queries, ensuring understanding of the study's purpose. They were informed that responses were subjective, with no right or wrong answers, and were assured of the questionnaire's anonymity and confidentiality. To address potential common method bias, the survey was conducted in three stages, each a month apart (September 1 –November 30, 2023), with each participant receiving a reward of RMB 5 upon completion.

1. On Time 1, the Family economic stress of college students was investigated, 900 questionnaires were sent out, and 820 questionnaires were collected, with a wastage rate of 8.9 per cent;

2. On Time 2, the love of money, perceived teacher support and effective part-time behavior of college students were investigated. 820 questionnaires were sent out and 750 questionnaires were collected, with a wastage rate of 8.0 per cent;

3. On Time 3, 750 questionnaires about career exploration were distributed and 690 questionnaires were regained, the wastage rate of the questionnaire was 8.5 per cent.

Additionally, with participants' consent, the research team included common control variables like gender, age, educational grade, and number of weekly courses [62, 63]. After survey completion, we used student IDs to match the three sets of questionnaires, discarding incomplete or invalid responses, yielding 600 valid questionnaires. The effective recovery rate was 66.67%. Among the participants, 341 were male (56.8%), and 259 were female (43.2%). The majority, 575 students (95.8%), were juniors, with an average age of 18.51 years and an average of 26.28 courses weekly.

## Ethics approval and consent to participate

All participants provided written informed consent for the collection of information and for the publication of data generated by the study. The target population for this study consists of adults over the age of 18. All participants voluntarily partake in the survey and may withdraw at any time. Considering that the survey content does not involve any sensitive issues and that the data collected are completely anonymous, according to Jinhua Open University, this study does not require ethical review. Furthermore, the research does not pose a risk to the physical or psychological health of participants, and the content of the survey will be conducted in a manner that respects and protects the privacy of participants.

## Measures

To ensure the reliability and validity of this survey, we used a widely validated scale. These scales are widely recognized in the research community and are effective in measuring the variables of concern in our study. Before the survey began, the questionnaire distribution team used a professional translator-back translation program to translate the English scale into Chinese, and proofread it several times to ensure the accuracy of the translation [64]. This process is essential to ensure the adaptability and accuracy of the questionnaire in different cultural contexts. A 5-point Likert scale was used throughout the study without special instructions (a scale of 1 to 5 on the questionnaire representing "strongly disagree" to "strongly agree").

**Family economic stress.** Based on the Wadsworth and Compas [9] Economic stress scale, we used four items to measure family economic stress from four aspects: clothing, food, housing and transportation. For example, "My family does not have enough money to buy new clothes," "My family does not have enough money to buy the food I like," "My family does not have enough money to buy a good house," "My family does not have money left over for entertainment." Participants were asked to report how often their families had experienced financial stress in the past year. A 5-point scale is used, where 1 means "never" and 5 means "always." In this study, the scale's Cronbach's alpha coefficient is 0.769.

**Love of money.** We adopted the love of money scale developed by Tang et al. [19]. The love of money scale included three subscales of "rich", "motivator" and "important", with a total of 9 questions. One sample item of "I'm going to be rich." In this study, the scale's Cronbach's alpha coefficient is 0.951.

**Teacher support.** We adopted the perceived teacher support scale developed by Patrick et al. [45]. The scale is a condensed version of the scale created by Ryan and Patrick [65] and modified by Patrick et al. [45]. We used eight items to measure perceived teacher support of students. Thus, as an indicator of perceived teacher support, higher scores indicate higher levels of perceived teacher support. One sample item of " My teacher understands how I feel about things." In this study, the scale's Cronbach's alpha coefficient is 0.930.

**Effective part-time behavior.** Based on the Mingqian and Sanman [66] Part-time behavior scale, we used four items to measure effective part-time behavior of students. For example,

"In general, the part-time job I choose is more related to my major" In this study, the scale's Cronbach's alpha coefficient is 0.960.

**Career exploration.** With a total of 17 items, the career exploration Scale developed by Stumpf et al. [7] was used to measure students' career exploration behaviors. Examples of questions include, " Reflected on how my past integrates with my future" and "I will engage in a variety of career activities." In this study, the scale's Cronbach's alpha coefficient is 0.891.

**Control variables.** According to previous studies, it is found that gender, age and education will affect individual career exploration [67]. At the same time, it is found that there are differences in career exploration behavior between courses weekly and support of college students [62, 63]. In order to validate the model more precisely, this study utilizes gender, age, grade, and number of weekly courses as control variables.

## Data analysis

In this study, we used SPSS 26.0 for Harman single factor test, descriptive analysis, correlation analysis and multiple linear regression analysis, while Amos 26.0 was used for confirmatory factor analysis. In addition, to estimate confidence intervals for mediating effects, we combined the three-step method of Baron and Kenny [68] with the Bootstrap method (using the PROCESS program) [69]. This study draws on the work of Edward and Lambert [70] to test the mediating effect with moderation, and adopts Bootstrap technology to compare the value and difference of the time connection effect when the level of moderator is high and low. The application of this method allows us to assess mediating effects more precisely and provide more detailed analysis where there are moderating effects.

## Results

### Common method deviation test

During the research and development phase, this study followed the multi-stage filling method recommended by Podsakoff et al. [71] to control for any possible common methodological biases at the methodological level. The data collected were subjected to Harman's single factor test at the data level of the survey results. The results showed that the variance explanation rate for the first factor was 24.89%, lower than the 40% criterion [72], suggesting no significant common methodology bias in the survey. In addition, the fitting index of confirmatory factor analysis of the single-factor model shown in Table 1 failed the test ($\chi^2$ = 20812.63, $df$ = 1593, RMSEA = 0.20, SRMR = 0.22, CFI = 0.28, TLI = 0.25). It is further confirmed that there is no significant common method bias between the study variables. These results provide strong support for the validity of the study.

**Table 1. Results of confirmatory factor analysis (N = 600).**

| Model | $\chi^2$ | $df$ | $\Delta\chi^2$ | RMSEA | SRMR | CFI | TLI |
|---|---|---|---|---|---|---|---|
| Five-factor model (hypothesis) | 6071.72 | 1584 | | .05 | .04 | .91 | .90 |
| Four-factor model (A+B) | 6861.30 | 1589 | 789.58*** | .06 | .08 | .86 | .83 |
| Three-factor model(B+C+D) | 13644.50 | 1592 | 7572.78*** | .16 | .14 | .51 | .48 |
| Two-factor model(A+B+C+D) | 14436.16 | 1594 | 8364.44*** | .17 | .15 | .47 | .45 |
| One factor model(A+B+C+D+E) | 20812.63 | 1593 | 14740.91*** | .20 | .22 | .28 | .25 |

*Note*. A: Family economic stress; B: Love of money; C: Teacher support; D: Effective part-time behavior; E: Career exploration; "+" means integration.

## Confirmatory factor analysis (CFA)

To evaluate the fit of the model in this study, we selected the following metrics: Chi-square differences must be significant, the root mean square of the approximation error (RMSEA) must be less than 0.08, and the comparative fit index (CFI) and Tuck-Lewis index (TLI) must be greater than 0.9. In this study, we compared multiple competing models and present the analysis results in Table 1. As shown in Table 1, the five-factor model in this study ($\chi^2$ = 6071.72, $df$ = 1584, RMSEA = 0.05, SRMR = 0.04, CFI = 0.91, TLI = 0.90) had better model adaptability than other competitive models. In addition, all the adaptability indexes of the five-factor model have passed the test. Therefore, this study concludes that all study variables are distinguishable on this basis. These results illustrate the validity of our selected model and provide a solid basis for further analysis.

**Correlation analysis.** Table 2 shows the results of correlation analysis between control variables and study variables. We found the significant positive correlation between family economic stress and career exploration ($r$ = 0.28, $p$ < 0.001), love of money and career exploration ($r$ = 0.15, $p$ < 0.001), which provides preliminary support for the positive prediction of family economic stress on career exploration.

**Hypothesis testing results.** *Results of direct effect tests.* As Model 4 of Table 3, it can be seen that there is a positive relationship between family economic stress and career exploration ($\beta$ = 0.14, $p$ < 0.001). Thus, Hypothesis 1 is supported.

*Results of mediating effect tests.* As Model 5 of Table 3, a significant positive relationship exists between family economic stress and career exploration ($\beta$ = 0.13, $p$ < 0.001), and a significant positive relationship exists between love of money and career exploration ($\beta$ = 0.07, $p$ < 0.01). The results above support love of money 's mediating role in the relationship between family economic stress and career exploration. To clarify this indirect effect once more, this study employs the Bootstrap method test [69]. Table 4 depicts the Bootstrap method test for the mediating effect. The direct and indirect effects of family economic stress and career exploration do not include zero within a confidence interval of 95%. Therefore, it can be demonstrated that love of money acts as a mediator between family economic stress and career exploration. Thus, Hypothesis 2 is supported.

*Results of moderating effect tests.* As shows Model 6 of Table 3, it can be seen revealed a significant positive relationship between the interaction between love of money and perceived teacher support on career exploration ($\beta$ = 0.10, $p$ < 0.05). At the same time, the Bootstrap test

**Table 2. Mean, standard deviation and correlation coefficient of variables (N = 600).**

| Variables | Mean | Standard Deviation | 1 | 2 | 3 | 4 | 5 | 6 | 7 | 8 | 9 |
|---|---|---|---|---|---|---|---|---|---|---|---|
| 1 Gender | 0.43 | 0.50 | | | | | | | | | |
| 2 Age | 18.51 | 0.79 | -.11* | | | | | | | | |
| 3 Grade | 1.05 | 0.22 | .05 | .28*** | | | | | | | |
| 4 Number of weekly courses | 26.28 | 7.33 | .11** | .02 | -.06 | | | | | | |
| 5 Family economic stress | 2.94 | 0.83 | .12** | -.06 | -.09* | .16*** | (0.769) | | | | |
| 6 Love of money | 3.83 | 0.72 | .01 | -.14*** | -.07 | .00 | .20*** | (0.951) | | | |
| 7 Effective part-time behavior | 3.36 | 0.80 | -.12** | -.03 | -.03 | -.28*** | -.09* | .20*** | (0.960) | | |
| 8 Teacher support | 3.64 | 0.66 | .02 | -.03 | -.07 | .01 | -.02 | .28*** | .54*** | (0.930) | |
| 9 Career exploration | 3.65 | 0.43 | .04 | -.04 | -.04 | .18*** | .28*** | .15*** | -.03 | .02 | (0.891) |

Note

***$p$<0.001

**$p$<0.01

* $p$<0.05 (Same below).

Table 3. Hypothesis testing model.

| Variables | Love of money | | Career exploration | | | | | | |
|---|---|---|---|---|---|---|---|---|---|
| | Model1 | Model 2 | Model 3 | Model 4 | Model 5 | Model 6 | Model 7 | Model 8 | Model 9 |
| **Control variable** | | | | | | | | | |
| Gender | -.01 | -.00 | .02 | -.01 | -.00 | .02 | .03 | .03 | .01 |
| Age | -.13** | -.12** | -.02 | -.01 | -.00 | .01 | -.00 | .00 | .00 |
| Grade | -.09 | -.04 | -.05 | -.01 | -.01 | -.08 | -.09 | -.10 | -.05 |
| Number of weekly courses | .01 | -.00 | .01*** | .01** | .01*** | .01*** | .01*** | .01*** | .01*** |
| **Independent variable** | | | | | | | | | |
| Family economic stress | | .17*** | | .14*** | .13*** | | . | | .12*** |
| **mediating variable** | | | | | | | | | |
| Love of money | | | | | .07** | .13*** | .10*** | .12*** | .08** |
| **moderating variable** | | | | | | | | | |
| Perceived teacher support | | | | | | .06* | | .06 | .05 |
| Effective part-time behavior | | | | | | | .03 | .02 | .02 |
| **Interaction item** | | | | | | | | | |
| Love of money * perceived teacher support | | | | | | .10* | | .05* | .02* |
| Love of money* effective part-time behavior | | | | | | | .09** | .07* | .06* |
| $R^2$ | .02 | .06 | .03 | .10 | .11 | .07 | .07 | .08 | .12 |
| $F$ | 3.29* | 7.42*** | 5.19*** | 12.83*** | 11.94*** | 7.86*** | 7.53*** | 5.44*** | 7.86*** |

of moderating effect is shown in Table 5. With low teacher support, the indirect effect of love of money on career exploration is lower with a 95% confidence interval (the effect value is 0.07). Love of money has a greater indirect impact on career exploration when there is a high degree of teacher support (the effect value is 0.20). To further elucidate the moderating effects, which are determined by the moderating level of moderating variables, the level of moderating variables is examined [72]. Fig 2 illustrates that the positive relationship between love of money and career exploration becomes stronger as the degree of teacher support increases [73]. Thus, Hypothesis 3 is supported.

As shows Model 7 of Table 3, it can be seen revealed a significant positive relationship between the interaction between love of money and effective part-time behavior on career exploration ($\beta = 0.09$, $p < 0.05$). At the same time, the Bootstrap test of moderating effect is shown in Table 6. With low effective part-time behavior, the indirect effect of love of money on career exploration is no significant with a 95% confidence interval ([-0.04, 0.09]). Love of money has a greater indirect impact on career exploration when there is a high degree of effective part-time behavior (the effect value is 0.17). To further elucidate the moderating effects, which are determined by the moderating level of moderating variables, the level of moderating variables is examined [72]. Fig 3 illustrates that the positive relationship between love of money and career exploration becomes stronger as the degree of effective part-time behavior increases [73]. Thus, Hypothesis 4 is supported.

Table 4. Bootstrap test of mediator.

| Mediating effect | Effect value | SE | 95% CI | |
|---|---|---|---|---|
| | | | Lower limit | Upper limit |
| Indirect effect | 0.01 | 0.004 | .00 | .02 |
| Direct effect | 0.13 | 0.022 | .08 | .17 |

*Note*. Bootstrap Sample Size N = 5000 (Same below)

**Table 5. Bootstrap test for the moderator of teacher support.**

| Moderating effect | Effect value | SE | 95% CI | |
|---|---|---|---|---|
| | | | Lower limit | Upper limit |
| LOW(-1SD) | 0.07 | 0.03 | .01 | .12 |
| Medium | 0.13 | 0.03 | .08 | .18 |
| HIGH(+1SD) | 0.20 | 0.04 | .11 | .28 |

To investigate the dual moderating effects of perceived teacher support and effective part-time behavior. Through Model 8 of Table 3, it can be seen that there is a significant positive relationship between the interaction between love of money and perceived teacher support and career exploration ($\beta$ = 0.05, $p$ < 0.05). Similarly, there was a significant positive correlation between love of money, effective part-time behavior, and career exploration ($\beta$ = 0.07, $p$ < 0.05). In this study, the Bootstrap method was used to test [69] to explore the double moderating effects. According to Table 7, the 95% confidence interval for the indirect effect of love of money on career exploration was no significant when both teacher support and effective part-time behavior were low (95%CI [-0.04, 0.10]). When the teacher support was low and effective part-time behavior was high, the indirect effect of love of money on career exploration

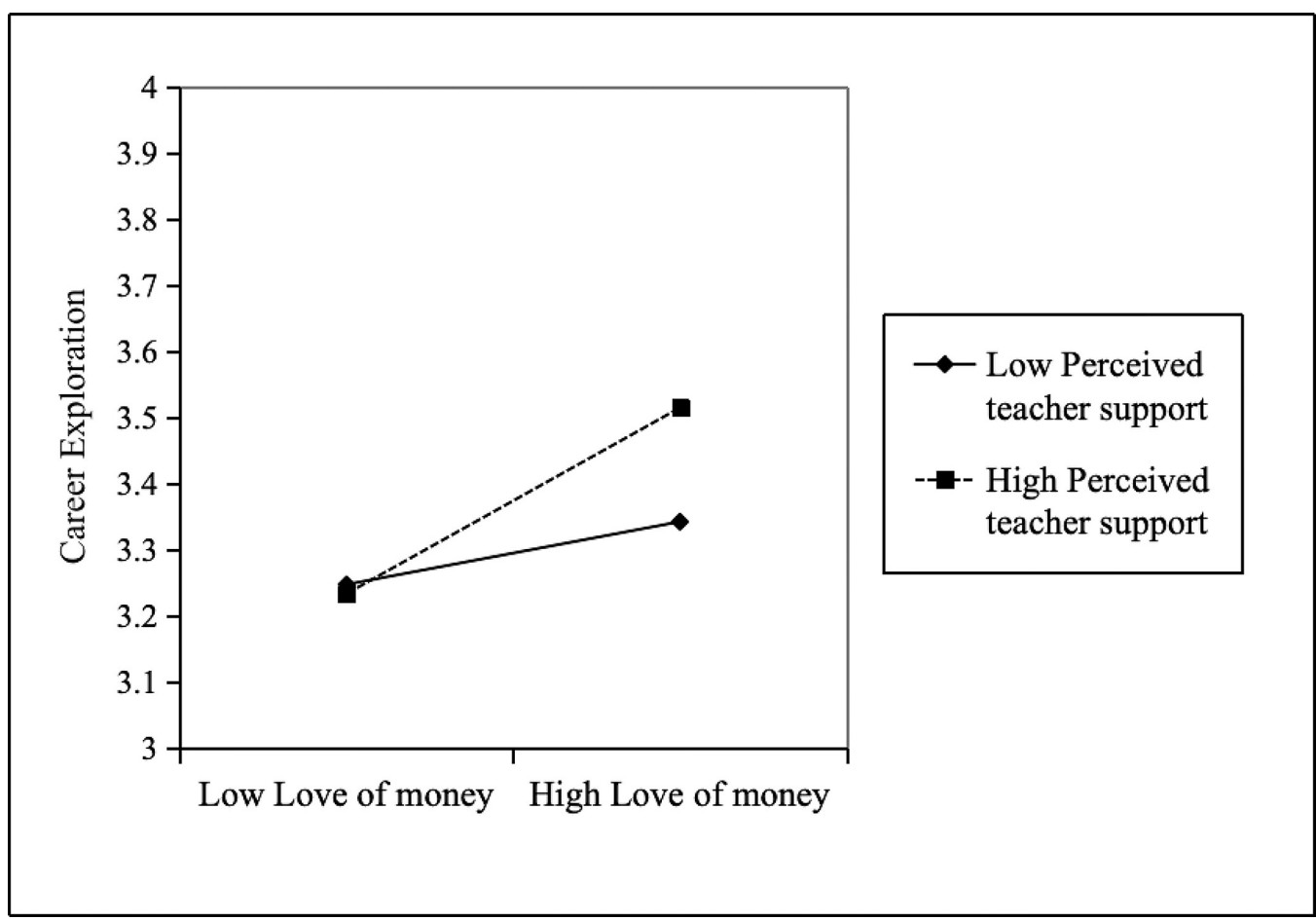

**Fig 2. Perceived teacher support as moderator between love of money and career exploration.**

**Table 6. Bootstrap test for the moderator of effective part-time behavior.**

| Moderating effect | Effect value | SE | 95% CI | |
|---|---|---|---|---|
| | | | Lower limit | Upper limit |
| Low(-1SD) | 0.03 | 0.03 | -.04 | .09 |
| Medium | 0.10 | 0.02 | .05 | .15 |
| High(+1SD) | 0.17 | 0.04 | .10 | .24 |

was increased (the effect values were 0.14). When both teacher support and effective part-time behavior were high, the indirect effect of love of money on career exploration is higher (the effect value is 0.21) [74]). To further clarify this double moderating effect, the study determined the level of moderating variables using Aiken et al. (1991)'s methodology [72]. Fig 4 illustrates that the positive relationship between love of money and career exploration is stronger the more teacher support and effective part-time behavior [75].

*Results of the moderated mediation model.* In this study, Bootstrap method was used to test the effect value of indirect effect under high and low level moderate variables [70]. From Table 8, it shows that the indirect effect of family economic stress on career exploration

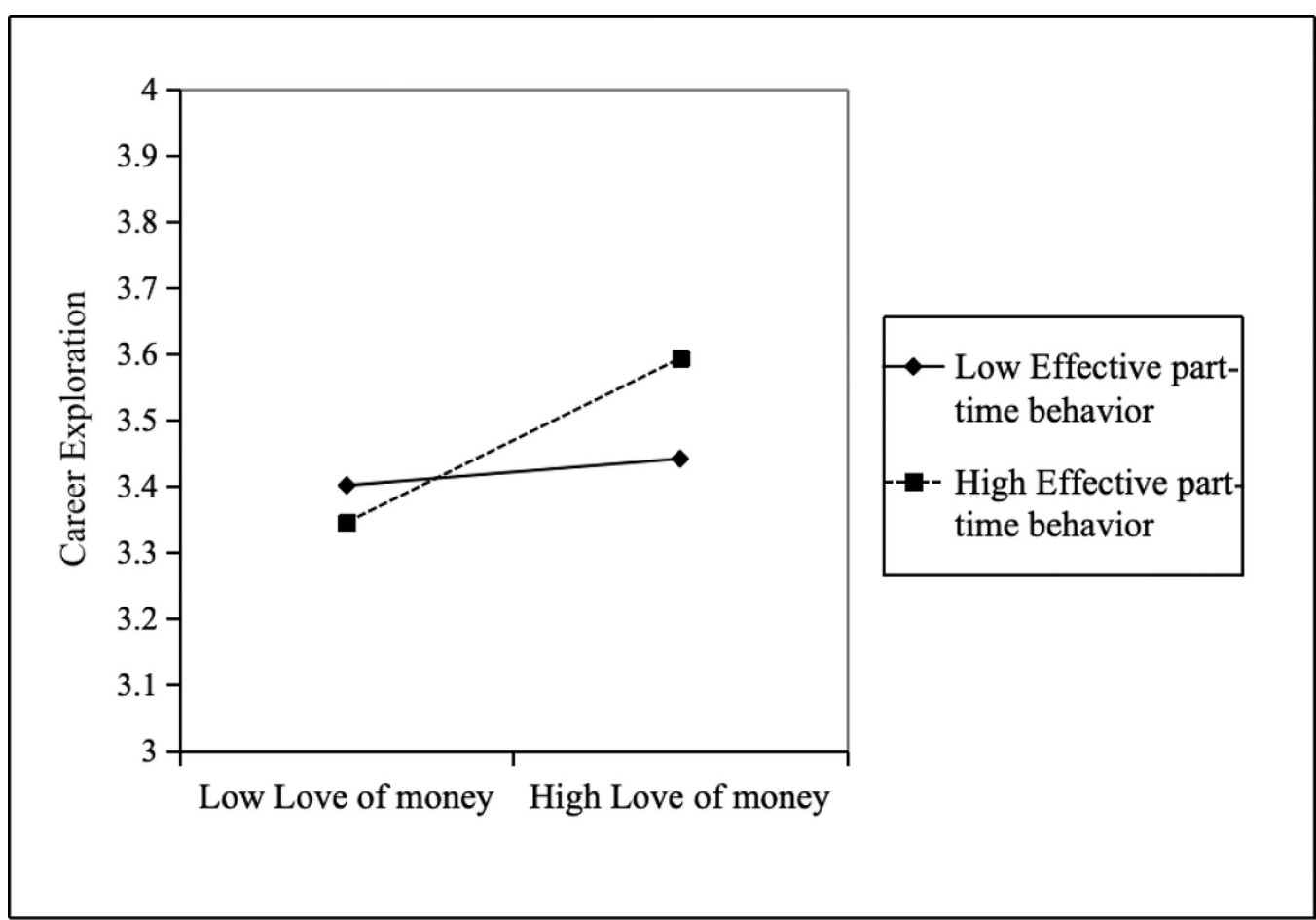

**Fig 3. Effective part-time behavior as moderator between love of money and career exploration.**

**Table 7. Bootstrap test for dual moderator between teacher support and effective part-time behavior.**

| Moderator 1(perceived teacher support) | Moderator 2(effective part-time behavior) | Effect value | SE | 95% CI | |
|---|---|---|---|---|---|
| | | | | Lower limit | Upper limit |
| Low(-1SD) | Low(-1SD) | 0.03 | 0.03 | -.04 | .10 |
| Low(-1SD) | High(+1SD) | 0.14 | 0.05 | .04 | .23 |
| High(+1SD) | Low(-1SD) | 0.10 | 0.06 | -.02 | .23 |
| High(+1SD) | High(+1SD) | 0.21 | 0.04 | .12 | .30 |

through love of money was 0.03 with high degree of teacher support and high level of effective part-time behavior, and its confidence interval is 95% [0.01, 0.04]. With low teacher support and low level of effective part-time behavior, the indirect effect o was 0.01, no significant, and its confidence interval was 95% [-0.01, 0.02]. When the level of teacher support and effective part-time behavior was inconsistent, the indirect effects were no significant or efficient. Therefore, the higher the perceived level of teacher and effective part-time behavior, the greater the indirect effect of family economic stress on career exploration via love of money. Thus, Hypothesis 5 is supported. All hypotheses are supported, as demonstrated in Table 9.

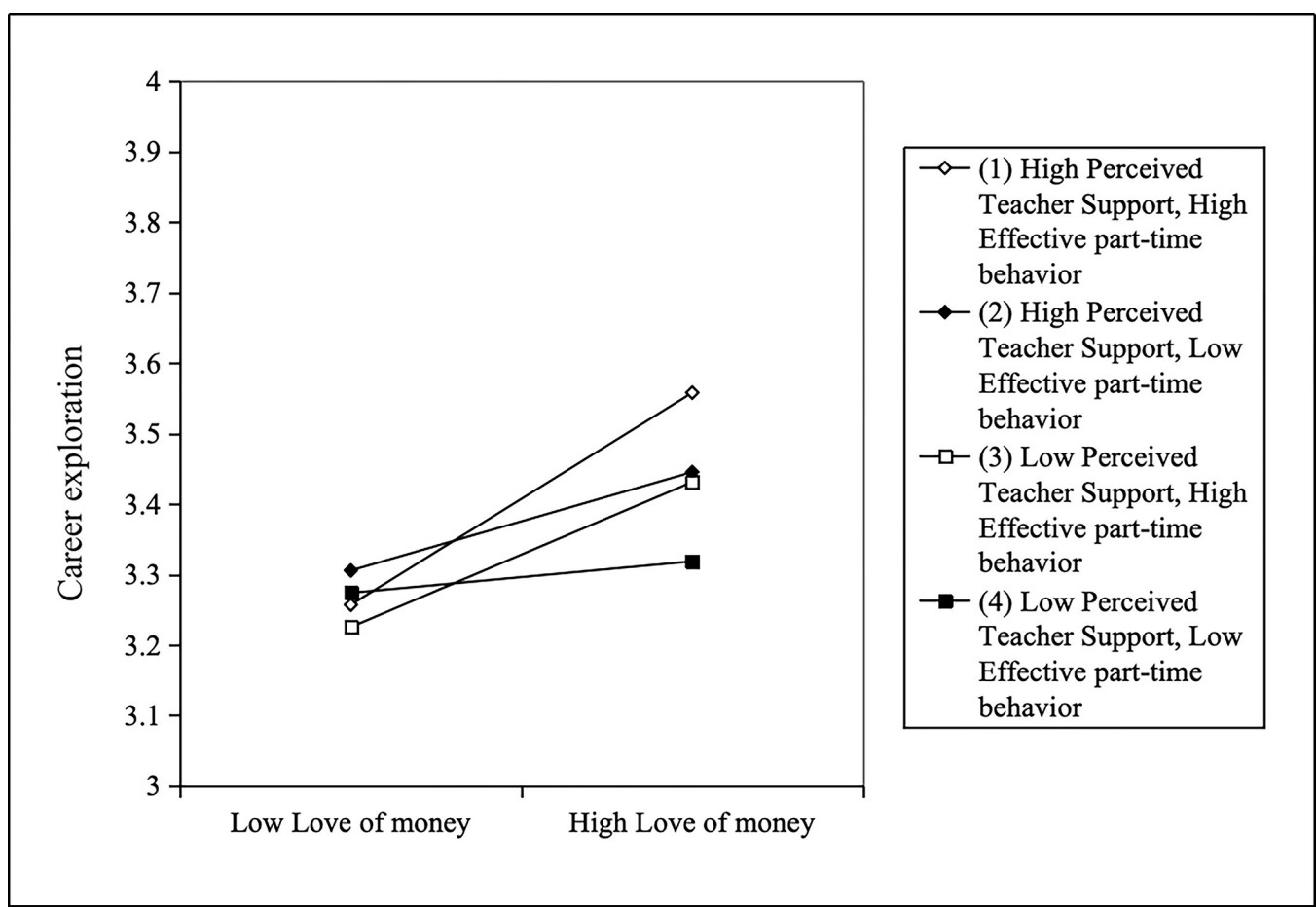

**Fig 4. Perceived teacher support and effective part-time behavior as dual moderators between love of money and career exploration.**

**Table 8. Bootstrap test with moderated mediating effect.**

| Mediator | Moderator 1(perceived teacher support) | Moderator 2(effective part-time behavior) | Indirect effect | SE | 95% CI | |
|---|---|---|---|---|---|---|
| | | | | | Lower limit | Upper limit |
| Love of money | Low(-1SD) | Low(-1SD) | 0.01 | 0.01 | -.01 | .01 |
| | Low(-1SD) | High(+1SD) | 0.02 | 0.01 | .01 | .03 |
| | High(+1SD) | Low(-1SD) | 0.01 | 0.01 | -.01 | .03 |
| | High(+1SD) | High(+1SD) | 0.03 | 0.01 | .01 | .04 |

## Discussion

### Conclusion

Understanding the factors influencing career choices and employment among vocational students is essential for developing effective career guidance and support strategies. The process of career exploration, a pivotal phase in shaping students' future professional paths, is influenced by various personal and environmental factors. This study, situated within the framework of social cognitive career theory (SCCT), aims to delve into the intricate relationship between family economic stress and career exploration, particularly among vocational school students. This study elucidates the complex relationship between family economic stress and career exploration among vocational students, emphasizing the mediating role of love of money and the moderating effects of teacher support and effective part-time behavior. The research offers critical insights for educators, policymakers, and career counselors, emphasizing the importance of understanding how economic background shapes career decisions. This knowledge is vital for developing customized career guidance programs for vocational students in China, aimed at effectively addressing their unique challenges and aspirations.

### Theoretical implications

First, this study enriches the social cognitive career theory by demonstrating how family economic stress influences career choices among vocational students, offering a more detailed understanding of the interplay between environmental factors and career development. Analogously, highlighting the importance of economic factors in shaping career trajectories [76], especially for students from lower socio-economic backgrounds.

Second, by identifying love of money as a mediating factor, the study offers a novel perspective on how financial attitudes shape career paths. This perspective underscores the profound influence of an individual's financial values and attitudes on their career choices [77, 78]. Essentially, it suggests that students who face economic challenges at home are likely to

**Table 9. Results of hypotheses.**

| Hypothesis Number | Hypothesis Description | Test Method | Conclusion |
|---|---|---|---|
| H1 | Family economic stress positively predicts vocational students' career exploration. | Regression Analysis | Supported |
| H2 | *Love of money mediates the relationship between family economic stress and career exploration.* | Mediated Regression Analysis | Supported |
| H3 | *Teacher support positively moderates the positive relationship between love of money and career exploration.* | Moderated Regression Analysis | Supported |
| H4 | *Effective part-time behavior positively moderates the positive relationship between love of money and career exploration.* | Moderated Regression Analysis | Supported |
| H5 | *Both vocational student perceived teacher support and effective part- time behavior positively moderate the indirect effect of family economic stress on career exploration through love of money.* | Multiple Regression Analysis | Supported |

prioritize careers that promise financial stability and success. This focus on monetary gain as a driving force for career exploration is particularly salient in contexts where economic pressure is a significant part of students' lives [79, 80] The study's findings reveal that the pursuit of financial security and wealth can be a crucial determinant in shaping students' career paths, particularly when family economic conditions are constrained.

Third, the study's exploration of teacher support and effective part-time work behaviors introduces a nuanced understanding of career exploration under economic stress. It demonstrates how these factors can significantly alter the way students influenced by financial motivations approach their career decisions. Teacher support offers guidance and emotional reinforcement, enabling students to align their financial goals with suitable career paths [81]. Effective part-time work behaviors provide practical experience and financial insight, aiding students in making informed career choices that balance their financial aspirations and professional interests [82, 83]. This understanding enriches the complexity of career decision-making processes among students navigating economic challenges.

Final, our study provides insights into the specific cultural context of Chinese vocational students, highlighting how cultural and familial economic factors can shape career exploration and decision-making processes in a non-Western context [84]. This perspective is vital for comprehending the varied influences on career decisions beyond Western contexts, revealing how cultural nuances and economic realities uniquely mold students' career pathways and exploration strategies in China [85].

## Practical implications

From the perspective of family economic stress, college employment guidance and career construction of higher vocational college students, this study provides some practical management suggestions for the issue of "Children from the poor families seems grow up earlier, that is, whether or not family economic stress promote career exploration".

**Financial realities and career aspirations: Tailoring vocational advice for students.** First, this study highlights the importance of developing more detailed, personalized career guidance programs in vocational education settings. Understanding a student's specific economic background helps to provide more effective and personalized career counseling [28, 86, 87]. This means that educators and career counselors need to consider a student's family financial situation and design career guidance strategies accordingly. For example, more information on financial planning and career development for financially stressed students [88], as well as practical advice on how to align career interests with financial goals [89]. This approach can help students navigate their career path more effectively while taking into account their financial needs and career aspirations.

**Navigating career paths: Insights for educational policy.** Second, this study's findings offer valuable insights for educational policymakers, particularly in the context of vocational education. By understanding the specific challenges faced by students under economic stress, policymakers can develop more targeted support strategies. This could involve allocating resources to enhance financial aid programs [90], creating workshops focused on financial literacy and career planning [91], and offering personalized counseling services [92]. These measures can help students navigate their career paths more effectively, taking into account both their financial constraints and professional aspirations. Essentially, these insights enable policymakers to create a more supportive and tailored educational environment for vocational students, addressing their unique needs and promoting their overall career success.

**Navigating financial pressures in career planning: The role of educators.** For educators, understanding the financial backgrounds and motivations of students is crucial. This

awareness enables educators to tailor their support strategies more effectively. By recognizing the economic pressures students face, teachers can provide guidance that aligns career aspirations with financial realities [92]. This approach involves introducing career options that offer financial stability, advising on financial planning, and fostering skills that bridge economic needs with professional interests [43]. For example, teachers can facilitate workshops or discussions that explore the intersection of finance and career goals, thus helping students navigate their career choices more holistically [93] Such support empowers students to make informed decisions, balancing their financial goals with their career objectives [94]. In addition, by introducing courses that combine career and financial planning, teachers help students to comprehensively consider economic and career interests in career choice, organize lectures and seminars to explore the relationship between career choice and financial management, and help students make informed career decisions.

**Practical learning: The impact of part-time jobs on vocational training.** Part-time work experiences provide students with practical insights into various career paths, helping them understand how different roles align with their financial goals [66]. This practical exposure enhances students' financial literacy and work readiness, bridging the gap between academic learning and real-world applications [95]. It also allows students to develop relevant skills and a deeper understanding of the financial implications of career choices, preparing them for a smoother transition into the workforce [96]. This approach aligns educational objectives with practical, financial outcomes, making vocational education more responsive to the real-world needs of students [97]. For example, schools should encourage and assist students to find part-time jobs related to their career goals, and provide appropriate support and guidance to ensure that students can balance their studies and work [98, 99], and achieve a balance between career goals and financial needs.

## Limitation and future research directions

Several limitations that point to future research opportunities are noteworthy. First, this study's reliance on self-reported data from a specific demographic (vocational students in China) may limit generalizability. The study's cross-sectional design also prevents causal inferences, and a narrow focus on monetary aspects of career exploration might overlook other crucial factors. Future research should consider a broader demographic, including students from various backgrounds and regions [100]. Longitudinal designs are recommended to establish causal relationships. Secondly, exploring other motivational factors like personal values or social influences would provide a more complete understanding of career exploration [101, 102]. Integrating qualitative methods would offer deeper insights into students' experiences with economic stress and career choices. Finally, since the study only included vocational students, the universality of its findings for undergraduates, postgraduates, and doctoral students remains uncertain. Future studies should include students from different educational stages to enhance the research's applicability.

## Supporting information

**S1 Data.**
(XLSX)

## Author Contributions

**Conceptualization:** Xiao-li Cheng, Guang-ya Ma, Lu-yao Zhang.

**Data curation:** Guang-ya Ma, Lei Lu.

**Formal analysis:** Guang-ya Ma, Lei Lu.

**Methodology:** Lu-yao Zhang, Lei Lu.

**Project administration:** Xiao-li Cheng.

**Resources:** Xiao-li Cheng, Lu-yao Zhang.

**Software:** Xiao-li Cheng.

**Supervision:** Xiao-li Cheng.

**Writing – original draft:** Xiao-li Cheng.

**Writing – review & editing:** Xiao-li Cheng, Lu-yao Zhang.

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
