## [Decision Letter · Decision Letter 0]

7 Jun 2024

PONE-D-24-15221Children from the Poor seems Grow up Earlier? The Mediating Role of Love for Money in Linking Family Economy Stress to Career Exploration: A Moderated Mediation ModelPLOS ONE

Dear Dr. Lu,

Thank you for submitting your manuscript to PLOS ONE. After careful consideration, we feel that it has merit but does not fully meet PLOS ONE’s publication criteria as it currently stands. Therefore, we invite you to submit a revised version of the manuscript that addresses the points raised during the review process.

We look forward to receiving your revised manuscript.

Kind regards,

Bo Pu, Ph.D.

Academic Editor

PLOS ONE

2. In this instance it seems there may be acceptable restrictions in place that prevent the public sharing of your minimal data. However, in line with our goal of ensuring long-term data availability to all interested researchers, PLOS’ Data Policy states that authors cannot be the sole named individuals responsible for ensuring data access (http://journals.plos.org/plosone/s/data-availability#loc-acceptable-data-sharing-methods).

Additional Editor Comments:

this manuscript should be improved.

Reviewers' comments:

Reviewer's Responses to Questions

**Comments to the Author**

1. Is the manuscript technically sound, and do the data support the conclusions?

Reviewer #1: Partly

Reviewer #2: Yes

Reviewer #3: Partly

2. Has the statistical analysis been performed appropriately and rigorously? 

Reviewer #1: No

Reviewer #2: Yes

Reviewer #3: Yes

3. Have the authors made all data underlying the findings in their manuscript fully available?

Reviewer #1: No

Reviewer #2: Yes

Reviewer #3: Yes

4. Is the manuscript presented in an intelligible fashion and written in standard English?

Reviewer #1: No

Reviewer #2: Yes

Reviewer #3: Yes

5. Review Comments to the Author

Reviewer #1: Thank you for the opportunity to review the paper entitled, “Children from the Poor seems Grow up Earlier? The Mediating Role of Love for Money in Linking Family Economy Stress to Career Exploration: A Moderated Mediation Model.” I recommend you to carefully review again and revise format throughout the manuscript to follow APA 7th Style. Though I did not list all the format errors, please carefully consider these for your revisions (e.g., past tense).

Reviewer #2: Dear Authors,

After reading your manuscript, a series of minor issues need to be addressed.

1. Please re-adjust your title in accordance to your results

2. please make a clearer depiction of Introduction vs. your literature review

3. please add some literature cited thresholds when discussing your results

4. as a suggestion, maybe add a table format where you explain the status of your hypotheses, for a clearer view

The text is well written and the results are adequately presented and referenced. I recommend the current manuscript for publication, after minor revisions.

Reviewer #3: Abstract

The opening sentence in the abstract has no connection and has to be improved.

Introduction

the gap in using love for money is confusing and not properly addressed. The authors should justify why they want to use money worship as a mediator based on either theory or previous studies.

Hypotheses development

What is the direction of the hypotheses you have set especially the intervening variables?

Methods

Justify why a 66.67% response rate can be used to generalize when you retrieved lower than your minimum sample size. Which criteria did you use to select participants for the survey?

Consider the choice of a bipolar scale as it is difficult to add positive to negative. That is adding SD to SA is not scientific. I suggest authors read on unipolar scales.

Why did the researchers introduce control in the methods without arguing for them in the introduction or conceptual framework?

Discussion

The researchers mentioned discussions but failed to present anything under them.

More concrete examples of how the findings could be implemented in practice would be beneficial.

The section could include more discussion on the practical significance of the findings, not just statistical significance.

6. PLOS authors have the option to publish the peer review history of their article (what does this mean?). If published, this will include your full peer review and any attached files.

Reviewer #1: No

Reviewer #2: No

Reviewer #3: **Yes: **Richard Kofi Boateng

---

## [Author Response · Author response to Decision Letter 0]

10 Jul 2024

Thank you for your valuable feedback. We have modified the manuscript according to your opinions and answered the questions raised by reviewers point by point all with the blue context like here. It is appreciated to be recognized by reviewers.

Thanks to the reviewer and editor's suggestion, we have done our best to revise the manuscript carefully. We hope that we succeeded in addressing your concerns and that you will now consider the paper suitable for publication in Plos One.

---

## [Decision Letter · Decision Letter 1]

23 Jul 2024

PONE-D-24-15221R1Children from the Poor Families Seem to Grow Up Earlier: An Examination of How Family Economy Stress Links to Career ExplorationPLOS ONE

Dear Dr. Lu,

Thank you for submitting your manuscript to PLOS ONE. After careful consideration, we feel that it has merit but does not fully meet PLOS ONE’s publication criteria as it currently stands. Therefore, we invite you to submit a revised version of the manuscript that addresses the points raised during the review process.

We look forward to receiving your revised manuscript.

Kind regards,

Bo Pu, Ph.D.

Academic Editor

PLOS ONE

Journal Requirements:

**Additional Editor Comments:**

Please revise your manuscript.

Reviewers' comments:

Reviewer's Responses to Questions

**Comments to the Author**

1. If the authors have adequately addressed your comments raised in a previous round of review and you feel that this manuscript is now acceptable for publication, you may indicate that here to bypass the “Comments to the Author” section, enter your conflict of interest statement in the “Confidential to Editor” section, and submit your "Accept" recommendation.

Reviewer #2: (No Response)

Reviewer #3: All comments have been addressed

2. Is the manuscript technically sound, and do the data support the conclusions?

Reviewer #2: Yes

Reviewer #3: Yes

3. Has the statistical analysis been performed appropriately and rigorously? 

Reviewer #2: No

Reviewer #3: Yes

4. Have the authors made all data underlying the findings in their manuscript fully available?

Reviewer #2: Yes

Reviewer #3: Yes

5. Is the manuscript presented in an intelligible fashion and written in standard English?

Reviewer #2: Yes

Reviewer #3: Yes

6. Review Comments to the Author

Reviewer #2: Dear Authors,

The Manuscript is well written.

However, there are several issues that need to urgently be addressed.

1. The Analysis should be backed up with the generally accepted thresholds and the appropriate citations for those limits.

2. The results should display what exactly happened with your hypotheses, I suggest a Table of a suggestive way to depict this information from the manuscript.

3. The results need and adequate discussion in the light of previous literature findings.

4. In regard with the Methodological approach, it should include:

- participants and procedure;

- measures that you used;

- analysis strategy;

- results where you need to enclose descriptive statistics and hypotheses testing.

5. Have you performed a pilot study? please motivate your answer.

6. It is not clear which instruments have you been using and who designed it? Why have you been using a 5 point Likert scale instead of a 7 point?

Best regards,

Reviewer #3: all comments have been duly attended to, however, the justification on why the authors did not add an argument for the control variables in the background and the framework has not been attended to enough. the authors should look at this comment and clarify it.

7. PLOS authors have the option to publish the peer review history of their article (what does this mean?). If published, this will include your full peer review and any attached files.

Reviewer #2: No

Reviewer #3: No

---

## [Author Response · Author response to Decision Letter 1]

29 Jul 2024

Thanks to the reviewer and editor's suggestion, we have done our best to revise the manuscript carefully. We hope that we succeeded in addressing your concerns and that you will now consider the paper suitable for publication in Plos One.

---

## [Editor Report · Decision Letter 2]

8 Sep 2024

Children from the Poor Families Seem to Grow Up Earlier: An Examination of How Family Economy Stress Links to Career Exploration

PONE-D-24-15221R2

Dear Dr. Lu,

We’re pleased to inform you that your manuscript has been judged scientifically suitable for publication and will be formally accepted for publication once it meets all outstanding technical requirements.

Kind regards,

Bo Pu, Ph.D.

Academic Editor

PLOS ONE

Additional Editor Comments (optional):

Thank you for improving this manuscript.
---

## [Editor Report · Acceptance letter]

1 Oct 2024

PONE-D-24-15221R2 

PLOS ONE

Dear Dr. Lu, 

I'm pleased to inform you that your manuscript has been deemed suitable for publication in PLOS ONE. Congratulations! Your manuscript is now being handed over to our production team.

Kind regards, 

on behalf of

Dr. Bo Pu 

Academic Editor

PLOS ONE